# Adaptive Batch Size for Privately Finding Second-Order Stationary Points

**Daogao Liu**[*]
University of Washington

**Kunal Talwar** [†]
Apple

## Abstract

There is a gap between finding a first-order stationary point (FOSP) and a second-order stationary point (SOSP) under differential privacy constraints, and it remains unclear whether privately finding an SOSP is more challenging than finding an FOSP. Specifically, Ganesh et al. (2023) claimed that an $\alpha$-SOSP can be found with $\alpha = \tilde{O}(\frac{1}{n^{1/3}} + (\frac{\sqrt{d}}{n\varepsilon})^{3/7})$, where $n$ is the dataset size, $d$ is the dimension, and $\varepsilon$ is the differential privacy parameter. However, a recent analysis revealed an issue in their saddle point escape procedure, leading to weaker guarantees. Building on the SpiderBoost algorithm framework, we propose a new approach that uses adaptive batch sizes and incorporates the binary tree mechanism. Our method not only corrects this issue but also improves the results for privately finding an SOSP, achieving $\alpha = \tilde{O}(\frac{1}{n^{1/3}} + (\frac{\sqrt{d}}{n\varepsilon})^{1/2})$. This improved bound matches the state-of-the-art for finding a FOSP, suggesting that privately finding an SOSP may be achievable at no additional cost.

## 1 Introduction

Privacy concerns have gained increasing attention with the rapid development of artificial intelligence and modern machine learning, particularly the widespread success of large language models. Differential privacy (DP) has become the standard notion of privacy in machine learning since it was introduced by Dwork et al. (2006). Given two neighboring datasets, $\mathcal{D}$ and $\mathcal{D}'$, differing by a single item, a mechanism $\mathcal{M}$ is said to be $(\varepsilon, \delta)$-differentially private if, for any event $\mathcal{X}$, it holds that:

$$\Pr[\mathcal{M}(\mathcal{D}) \in \mathcal{X}] \leq e^{\varepsilon} \Pr[\mathcal{M}(\mathcal{D}') \in \mathcal{X}] + \delta.$$

In this work, we focus on the stochastic optimization problem under the constraint of DP. The loss function is defined below:

$$F_{\mathcal{P}}(x) := \mathbb{E}_{z \sim \mathcal{P}} f(x; z),$$

where the functions may be non-convex, the underlying distribution $\mathcal{P}$ is unknown, and we are given a dataset $\mathcal{D} = \{z_i\}_{i \in [n]}$ drawn i.i.d. from $\mathcal{P}$. Notably, our goal is to design a private algorithm with provable utility guarantees under the i.i.d. assumption.

Minimizing non-convex functions is generally challenging and often intractable, but most models used in practice are not guaranteed to be convex. How, then, can we explain the success of optimization methods in practice? One possible explanation is the effectiveness of Stochastic Gradient Descent (SGD), which is well-known to be able to find an $\alpha$-first-order stationary point (FOSP) of a non-convex function $f$—that is, a point $x$ such that $\|\nabla f(x)\| \leq \alpha$—within $O(1/\alpha^2)$ steps (Nesterov, 1998). However, FOSPs can include saddle points or even local maxima. Thus, we focus on finding second-order stationary points (SOSP), for the non-convex function $F_{\mathcal{P}}$.

Non-convex optimization has been extensively studied in recent years due to its central role in modern machine learning, and we now have a solid understanding of the complexity involved in finding FOSPs and SOSPs (Ghadimi & Lan, 2013; Agarwal et al., 2017; Carmon et al., 2020; Zhang et al.,

---

[*]work was done while interning at Apple, Email: `liudaogao@gmail.com`
[†]Email: `kunal@kunaltalwar.org`

2020). Variance reduction techniques have been shown to improve the theoretical complexity, leading to the development of several promising algorithms such as Spider (Fang et al., 2018), SARAH (Nguyen et al., 2017), and SpiderBoost (Wang et al., 2019b). More recently, private non-convex optimization has emerged as an active area of research (Wang et al., 2019a; Tran & Cutkosky, 2022; Arora et al., 2023; Gao & Wright, 2023; Ganesh et al., 2023; Wang et al., 2023; Lowy et al., 2024; Kornowski et al., 2024; Menart et al., 2024).

## 1.1 OUR MAIN RESULT

In this work, we study how to find the SOSP of $F_{\mathcal{P}}$ privately. Let us formally define the FOSP and the SOSP. For more on Hessian Lipschitz continuity and related background, see the preliminaries in Section 2.

**Definition 1.1** (FOSP). *For $\alpha \geq 0$, we say a point $x$ is an $\alpha$-first-order stationary point ($\alpha$-FOSP) of a function $g$ if $\|\nabla g(x)\| \leq \alpha$.*

**Definition 1.2** (SOSP, Nesterov & Polyak (2006); Agarwal et al. (2017)). *For a function $g : \mathbb{R}^d \to \mathbb{R}$ which is $\rho$-Hessian Lipschitz, we say a point $x \in \mathbb{R}^d$ is $\alpha$-second-order stationary point ($\alpha$-SOSP) of $g$ if $\|\nabla g(x)\|_2 \leq \alpha \bigwedge \nabla^2 g(x) \succeq -\sqrt{\rho\alpha}I_d$.*

Given the dataset size of $n$, privacy parameters $\varepsilon, \delta$, and functions defined over $d$-dimensional space, Ganesh et al. (2023) proposed a private algorithm that can find an $\alpha_S$-SOSP for $F_{\mathcal{P}}$ with

$$\alpha_S = \tilde{O}\big(\frac{1}{n^{1/3}} + (\frac{\sqrt{d}}{n\varepsilon})^{3/7}\big).$$

However, as shown in Arora et al. (2023), the state-of-the-art bound for privately finding an $\alpha_F$-FOSP is tighter: [1]

$$\alpha_F = \tilde{O}\big(\frac{1}{n^{1/3}} + (\frac{\sqrt{d}}{n\varepsilon})^{1/2}\big)$$

When the privacy parameter $\varepsilon$ is sufficiently small, and the error term depending on it dominates the non-private term $1/n^{1/3}$, we observe that $\alpha_F \ll \alpha_S$. This raises the question: is finding an SOSP under differential privacy constraints more difficult than finding an FOSP?

Moreover, as pointed out by Tao et al. (2025), the results in Ganesh et al. (2023) may be overly optimistic. In particular, the saddle point escape subprocedure used in Ganesh et al. (2023) was designed for the full-batch setting. However, to mitigate dependence issues, the algorithm performs only a single pass over the $n$ functions and relies on minibatches instead. A direct fix for this issue leads to a weaker guarantee on the minimum eigenvalue of the Hessian, specifically,

$$\nabla^2 F_{\mathcal{P}}(x) \succeq -\sqrt{\rho}d^{1/5}\alpha^{2/5}I_d,$$

which fails to satisfy Definition 1.2. To address this, Tao et al. (2025) proposed an alternative correction, but their method resulted in weaker theoretical guarantees than Ganesh et al. (2023) originally claimed, yielding

$$\alpha = \tilde{O}\big(\frac{1}{n^{1/3}} + (\frac{\sqrt{d}}{n\varepsilon})^{2/5}\big).$$

This work improves upon the results of Ganesh et al. (2023). In addition, we introduce a new saddle point escape subprocedure that correctly addresses the issue in Ganesh et al. (2023) while fully recovering—and even strengthening—the theoretical guarantees.

Specifically, we present an algorithm that finds an $\alpha$-SOSP with privacy guarantees, where:

$$\alpha = \tilde{O}(\alpha_F) = \tilde{O}\big(\frac{1}{n^{1/3}} + (\frac{\sqrt{d}}{n\varepsilon})^{1/2}\big).$$

This improved bound suggests that when we try to find the stationary point privately, we can find the SOSP for free under additional (standard) assumptions.

---

[1]As proposed by Lowy et al. (2024), allowing exponential running time enables the use of the exponential mechanism to find a warm start, which can further improve the bounds for both FOSP and SOSP.

It is also worth noting that, our improvement primarily affects terms dependent on the privacy parameters. In the non-private setting, as $\varepsilon \to \infty$ (i.e., without privacy constraints), all the results discussed above achieve a bound of $\tilde{O}(1/n^{1/3})$, which matches the non-private lower bound established by Arjevani et al. (2023) in high-dimensional settings (where $d \geq \tilde{\Omega}(1/\alpha^4)$). However, to our knowledge, whether this non-private term of $\tilde{O}(1/n^{1/3})$ can be further improved in low dimension remains an open question.

## 1.2 OVERVIEW OF TECHNIQUES

In this work, we build on the SpiderBoost algorithm framework, similar to prior approaches (Arora et al., 2023; Ganesh et al., 2023), to find second-order stationary points (SOSP) privately. At a high level, our method leverages two types of gradient oracles: $\mathcal{O}_1(x) \approx \nabla f(x)$, which estimates the gradient at point $x$, and $\mathcal{O}_2(x, y) \approx \nabla f(x) - \nabla f(y)$, which estimates the gradient difference between two points, $x$ and $y$. When performing gradient descent, to compute the gradient estimator $\nabla_t$ at point $x_t$, we can either use $\nabla_t = \mathcal{O}_1(x_t)$ for a direct estimate, or $\nabla_t = \nabla_{t-1} + \mathcal{O}_2(x_t, x_{t-1})$ to update based on the previous gradient. In our setting, $\mathcal{O}_1$ is more accurate but incurs higher computational or privacy costs.

The approach in Arora et al. (2023) adopts SpiderBoost by querying $\mathcal{O}_1$ periodically: they call $\mathcal{O}_1$ once and then use $\mathcal{O}_2$ for $q$ subsequent queries, controlled by a hyperparameter $q$. Their method ensures the gradient estimators are sufficiently accurate on average, which works well for finding first-order stationary points (FOSP). However, finding an SOSP, where greater precision is required, presents additional challenges when relying on average-accurate gradient estimators.

To address this, Ganesh et al. (2023) introduced a variable called $\mathrm{drift}_t := \sum_{i=t_0+1}^t \|x_i - x_{i-1}\|^2$, where $t_0$ is the index of the last iteration when $\mathcal{O}_1$ was queried. If $\mathrm{drift}_t$ remains small, the gradient estimator stays accurate enough, allowing further queries of $\mathcal{O}_2$. However, if $\mathrm{drift}_t$ grows large, the gradient estimator's accuracy deteriorates, signaling the need to query $\mathcal{O}_1$ for a fresh, more accurate estimate. This modification enables the algorithm to maintain the precision necessary for privately finding an SOSP.

Our improvement introduces two new components: the use of the tree mechanism instead of using the Gaussian mechanism as in Ganesh et al. (2023), and the implementation of adaptive batch sizes for constructing $\mathcal{O}_2$.

In the prior approach using the Gaussian mechanism, a noisy gradient estimator $\nabla_{t-1}$ is computed, and the next estimator is updated via $\nabla_t = \nabla_{t-1} + \mathcal{O}_2(x_t, x_{t-1}) + g_t$, where $g_t$ is Gaussian noise added to preserve privacy. Over multiple iterations, the accumulation of noise $\sum g_t$ can severely degrade the accuracy of the gradient estimator, requiring frequent re-queries of $\mathcal{O}_1$. On the other hand, the tree mechanism mitigates this issue when frequent queries to $\mathcal{O}_2$ are needed.

However, simply replacing the Gaussian mechanism with the tree mechanism and using a fixed batch size does not yield optimal results. In worst-case scenarios, where the function's gradients are large, the $\mathrm{drift}$ grows quickly, necessitating frequent calls to $\mathcal{O}_1$, which diminishes the advantages of the tree mechanism.

To address this, we introduce adaptive batch sizes. In Ganesh et al. (2023), the oracle $\mathcal{O}_2$ is constructed by drawing a fixed batch of size $B$ from the unused dataset and outputting $\mathcal{O}_2(x_t, x_{t-1}) := \sum_{z \in S_t} \frac{\nabla f(x_t;z) - \nabla f(x_{t-1};z)}{B}$. Given an upper bound on $\mathrm{drift}$, they guaranteed that $\|x_t - x_{t-1}\| \leq D$ for some parameter $D$, thereby bounding the sensitivity of $\mathcal{O}_2$.

In contrast, we dynamically adjust the batch size in proportion to $\|x_t - x_{t-1}\|$, setting $B_t \propto \|x_t - x_{t-1}\|$, and compute $\mathcal{O}_2(x_t, x_{t-1}) := \sum_{z \in S_t} \frac{\nabla f(x_t;z) - \nabla f(x_{t-1};z)}{B_t}$. Fixed batch sizes present two drawbacks: (i) when $\|x_t - x_{t-1}\|$ is large, the gradient estimator has higher sensitivity and variance, leading to worse estimate accuracy; (ii) when $\|x_t - x_{t-1}\|$ is small, progress in terms of function value decrease is limited. Using a fixed batch size forces us to handle both cases simultaneously: we must add noise and analyze accuracy assuming a worst-case large $\|x_t - x_{t-1}\|$, but for utility analysis, we pretend $\|x_t - x_{t-1}\|$ is small to examine the function value decrease. The adaptive batch size resolves this paradox: it allows us to control sensitivity and variance adaptively. When $\|x_t - x_{t-1}\|$ is small, we decrease the batch size but can still control the variance and sensitivity; when it is small, the function value decreases significantly, aiding in finding an SOSP.

By combining the tree mechanism with adaptive batch sizes, we improve the accuracy of gradient estimation and achieve better results for privately finding an SOSP.

**Fixing the Error:** Building upon our discussion of obtaining more accurate gradient estimations through adaptive batch sizes, we extend this approach to Hessian estimations, achieving accuracy in terms of the operator norm. Upon encountering a potential saddle point—characterized by a small gradient norm—we utilize the Hessian to inform the gradient estimation over several iterations. This process is analogous to performing stochastic power iteration methods on the Hessian to identify the direction corresponding to the smallest eigenvalue. If the Hessian exhibits a small enough eigenvalue (say smaller than $-\sqrt{\rho\alpha}$), we demonstrate that this approach facilitates a significant drift, effectively enabling successful escape from the saddle point. This idea can also be applied to Ganesh et al. (2023) to recover their claimed rate.

## 1.3 OTHER RELATED WORK

A significant body of literature on private optimization focuses on the convex setting, where it is typically assumed that each function $f(;z)$ is convex for any $z$ in the universe (e.g., (Chaudhuri et al., 2011; Bassily et al., 2014; 2019; Feldman et al., 2020; Asi et al., 2021; Kulkarni et al., 2021; Carmon et al., 2023; Gopi et al., 2023)).

The tree mechanism, originally introduced by the differential privacy (DP) community (Dwork et al., 2010; Chan et al., 2011) for the continual observation, has inspired tree-structure private optimization algorithms like Asi et al. (2021); Bassily et al. (2021); Arora et al. (2023); Zhang et al. (2024). One can also use the matrix mechanism Fichtenberger et al. (2023) which improves the tree mechanism by a constant factor. Some prior works have explored adaptive batch size techniques in optimization. For instance, De et al. (2016) introduced adaptive batch sizing for stochastic gradient descent (SGD), while Ji et al. (2020) combined adaptive batch sizing with variance reduction techniques to modify SVRG and Spider algorithms. However, these works' motivations and approaches to setting adaptive batch sizes differ from ours. To the best of our knowledge, we are the first to propose using adaptive batch sizes in the context of optimization under differential privacy constraints.

Most of the non-convex optimization literature assumes that the functions being optimized are smooth. Recent work has begun addressing non-smooth, non-convex functions as well, as seen in Zhang et al. (2020); Kornowski & Shamir (2021); Davis et al. (2022); Jordan et al. (2023).

## 2 PRELIMINARIES

Throughout the paper, we use $\|\cdot\|$ to represent both the $\ell_2$ norm of a vector and the operator norm of a matrix when there is no confusion.

**Definition 2.1** (Lipschitz, Smoothness and Hessian Lipschitz). *Let $\mathcal{K} \subseteq \mathbb{R}^d$. Given a twice differentiable function $f : \mathcal{K} \to \mathbb{R}$, we say $f$ is $G$-Lipschitz, if for all $x_1, x_2 \in \mathcal{K}$, $|f(x_1) - f(x_2)| \leq G\|x_1 - x_2\|$; we say $f$ is $M$-smooth, if for all $x_1, x_2 \in \mathcal{K}$, $\|\nabla f(x_1) - \nabla f(x_2)\| \leq M\|x_1 - x_2\|$, and we say the function $f$ is $\rho$-Hessian Lipschitz, if for all $x_1, x_2 \in \mathcal{K}$, we have $\|\nabla^2 f(x_1) - \nabla^2 f(x_2)\| \leq \rho\|x_1 - x_2\|$.*

## 2.1 OTHER TECHNIQUES

As mentioned in the introduction, we use the tree mechanism (Algorithm 1) to privatize the algorithm, whose formal guarantee is stated below:

**Theorem 2.2** (Tree Mechanism, Dwork et al. (2010); Chan et al. (2011)). *Let $\mathcal{Z}_1, \cdots, \mathcal{Z}_\Sigma$ be dataset spaces, and $\mathcal{X}$ be the state space. Let $\mathcal{M}_i : \mathcal{X}^{i-1} \times \mathcal{Z}_i \to \mathcal{X}$ be a sequence of algorithms for $i \in [\Sigma]$. Let $\mathcal{A} : \mathcal{Z}^{(1:\Sigma)} \to \mathcal{X}^\Sigma$ be the algorithm that given a dataset $Z_{1:\Sigma} \in \mathcal{Z}^{(1:\Sigma)}$, sequentially computes $X_i = \sum_{j=1}^{i} \mathcal{M}_j(X_{1:j-1}, Z_j) + \text{TREE}(i)$ for $i \in [\Sigma]$, and then outputs $X_{1:\Sigma}$.*

*Suppose for all $i \in [\Sigma]$, and neighboring $Z_{1:\Sigma}, Z'_{1:\Sigma} \in \mathcal{Z}^{(1:\Sigma)}, \|\mathcal{M}_i(X_{1:i-1}, Z_i) - \mathcal{M}_i(X_{1:i-1}, Z'_i)\| \leq s$ for all auxiliary inputs $X_{1:i-1} \in \mathcal{X}^{i-1}$. Then setting $\sigma = \frac{4s\sqrt{\log \Sigma \log(1/\delta)}}{\varepsilon}$,*

*Algorithm 1 is $(\varepsilon, \delta)$-DP. Furthermore, with probability at least $1 - \Sigma \cdot \iota$, for all $t \in [\Sigma]$ :* $\|\mathrm{TREE}(t)\| \lesssim \sqrt{d \log(1/\iota)}\sigma$.

---

**Algorithm 1** Tree Mechanism

---

1: **Input:** Noise parameter $\sigma_{\mathrm{tree}}$, sequence length $\Sigma$
2: Define $\mathcal{T} := \{(u,v) : u = j \cdot 2^{\ell-1} + 1, v = (j+1) \cdot 2^{\ell-1}, 1 \leq \ell \leq \log \Sigma, 0 \leq j \leq \Sigma/2^{\ell-1} - 1\}$
3: Sample and store $\zeta_{(u,v)} \sim \mathcal{N}(0, \sigma^2)$ for each $(u,v) \in \mathcal{T}$
4: **for** $t = 1, \cdots, \Sigma$ **do**
5:     Let $\mathrm{TREE}(t) \leftarrow \sum_{(u,v) \in \mathrm{NODE}(t)} \zeta_{(u,v)}$
6: **end for**
7: **Return:** $\mathrm{TREE}(t)$ for each $t \in [\Sigma]$

8: **Function NODE:**
9: **Input:** index $t \in [\Sigma]$
10: Initialize $S = \{\}$ and $k = 0$
11: **for** $i = 1, \cdots, \lceil \log \Sigma \rceil$ while $k < t$ **do**
12:     Set $k' = k + 2^{\lceil \log \Sigma \rceil - i}$
13:     **if** $k' \leq t$ **then**
14:         $S \leftarrow S \cup \{(k+1, k')\}, k \leftarrow k'$
15:     **end if**
16: **end for**

---

We also need the concentration inequality for norm-subGaussian random vectors.

**Definition 2.3** (SubGaussian, and Norm-SubGaussian)**.** *We say a random vector $x \in \mathbb{R}^d$ is Sub-Gaussian ($\mathrm{SG}(\zeta)$) if there exists a positive constant $\zeta$ such that $\mathbb{E}\, e^{\langle v, x - \mathbb{E}\, x \rangle} \leq e^{\|v\|^2 \zeta^2 / 2}, \ \forall v \in \mathbb{R}^d$. We say $x \in \mathbb{R}^d$ is norm-SubGaussian ($\mathrm{nSG}(\zeta)$) if there exists $\zeta$ such that $\Pr[\|x - \mathbb{E}\, x\| \geq t] \leq 2e^{-\frac{t^2}{2\zeta^2}}, \forall t \in \mathbb{R}$.*

**Lemma 2.4** (Hoeffding type inequality for norm-subGaussian, Jin et al. (2019))**.** *Let $x_1, \cdots, x_k \in \mathbb{R}^d$ be random vectors, and for each $i \in [k]$, $x_i \mid \mathcal{F}_{i-1}$ is zero-mean $\mathrm{nSG}(\zeta_i)$ where $\mathcal{F}_i$ is the corresponding filtration. Then there exists an absolute constant $c$ such that for any $\delta > 0$, with probability at least $1 - \omega$, $\|\sum_{i=1}^{k} x_i\| \leq c \cdot \sqrt{\sum_{i=1}^{k} \zeta_i^2 \log(2d/\omega)}$, which means $\sum_{i=1}^{k} x_i$ is $\mathrm{nSG}(\sqrt{c \log(d) \sum_{i=1}^{k} \zeta_i^2})$.*

**Theorem 2.5** (Matrix Azuma Inequality, Tropp (2012))**.** *Consider a finite sequence of random self-adjoint matrices $X_1, \cdots, X_n$ with common dimensions $d \times d$ and $\mathbb{E}[X_i \mid \mathcal{F}_{i-1}] = 0, X_i \preceq A_i$ a.s., $\forall i$. Let $\sigma^2 = \|\sum_{i=1}^{n} A_i^2\|$. Let $S = \sum_{i=1}^{n} X_i$. Then for any $t \geq 0$, we have*

$$\Pr[\|S\| \geq t] \leq d \cdot \exp(-\frac{-t^2}{8\sigma^2}).$$

## 3 SOSP

We make the following assumption for our main result.

**Assumption 3.1.** *Let $G, \rho, M, B > 0$. Any function drawn from $\mathcal{P}$ is $G$-Lipschitz, $\rho$-Hessian Lipschitz, and $M$-smooth, almost surely. Moreover, we are given a public initial point $x_0$ such that $F_{\mathcal{P}}(x_0) - \inf_x F_{\mathcal{P}}(x) \leq B$.*

We modify the Stochastic SpiderBoost used in Ganesh et al. (2023) and state it in Algorithm 3. The following standard lemma plays a crucial role in finding stationary points of smooth functions:

**Lemma 3.2.** *Assume $F$ is $M$-smooth and let $\eta = 1/M$. Let $x_{t+1} = x_t - \eta g_t$. Then we have $F(x_{t+1}) \leq F(x_t) + \eta \|g_t\| \cdot \|\nabla F(x_t) - g_t\| - \frac{\eta}{2}\|g_t\|^2$. Moreover, if $\|\nabla F(x_t)\| \geq \gamma$ and $\|g_t - \nabla F(x_t)\| \leq \gamma/4$, we have*

$$F(x_{t+1}) - F(x_t) \leq -\eta \|g_t\|^2 / 16.$$

*Proof.* By the assumption of the smoothness, we know

$$F(x_{t+1}) \leq F(x_t) + \langle \nabla F(x_t), x_{t+1} - x_t \rangle + \frac{M}{2} \|x_{t+1} - x_t\|^2$$

$$= F(x_t) - \eta/2\|g_t\|^2 - \langle \nabla F(x_t) - g_t, \eta g_t \rangle$$

$$\leq F(x_t) + \eta \|\nabla F(x_t) - g_t\| \cdot \|g_t\| - \frac{\eta}{2} \|g_t\|^2.$$

When $\|\nabla F(x_t)\| \geq \gamma$ and $\|g_t - \nabla F(x_t)\| \leq \gamma/4$, the conclusion follows from the calculation. $\square$

Lemma 3.2 shows that one can be able to find the FOSP with inexact gradient estimates as long as the estimated error is small enough. A key challenge in finding an SOSP, compared to an FOSP, is ensuring that the algorithm can effectively escape saddle points. Existing analyses of saddle point escape using inexact gradients are insufficient for our purposes. To address this, we propose a new approach that leverages both inexact gradients and Hessians for escaping saddle points. The details are presented in the following subsection.

## 3.1 Escape Saddle Point with Hessian

We present the subprocedure for escaping saddle points in this section, with its pseudocode provided in Algorithm 2. The core idea is straightforward: given a sufficiently accurate estimate of the function's Hessian, we can apply the power method to escape the saddle point. If the Hessian's smallest eigenvalue is sufficiently negative, then after a certain number of steps, the iterate will have moved a significant distance, resulting in a substantial decrease in function value—indicating a successful escape.

**Definition 3.3.** *We say the estimation $(g, H)$ of the pair of gradient and Hessian is $(\gamma, \varkappa)$-accurate at point $x$, if*

$$\|g - \nabla F(x)\|_2 \leq \gamma, \|H - \nabla^2 F(x)\|_2 \leq \varkappa.$$

---

**Algorithm 2** Escape from Saddle Point

1: **Input:** initial point $x_0$ such that $\|\nabla F(x)\| \leq \alpha$, $(\gamma, \varkappa)$-accurate estimation pair $(g, H)$ at $x_0$, parameters $\Gamma, \Xi$
2: **Process:**
3: $y \leftarrow x_0$
4: **for** $t = 1, \cdots, \Gamma$ **do**
5:     $g_{t-1} = g + H(x_{t-1} - x_0)) + \zeta_{t-1}$, where $\zeta_{t-1} \sim \mathcal{N}(0, \sigma_{t-1}^2 I_d)$.
6:     $x_t = x_{t-1} - \eta g_{t-1}$
7:     **if** $\|x_t - x_0\| \geq \Xi$ **then**
8:         $y \leftarrow x_t$
9:         **Break**
10:    **end if**
11: **end for**
12: **Output:** $y$

---

The Gaussian noise $\zeta_{t-1}$ added in Line 5 is for the privacy purpose. We have the following guarantee of Algorithm 2:

**Theorem 3.4.** *Suppose the function $F$ satisfies the Assumption 3.1, and the initial point is a saddle point such that $\|\nabla F(x)\| \leq \alpha$ and $\nabla^2 F(x) \succeq -\sqrt{\rho\alpha} I_d$. When $\alpha \geq \gamma \log^3(dBM/\rho\iota)$, $\sigma_t = \frac{\gamma}{\sqrt{d \log(T/\iota)}}$ $\Gamma = \tilde{O}(M/\sqrt{\rho\gamma})$, $\varkappa\Xi \leq \gamma, \eta = 1/M$ and $\Xi = \sqrt{\gamma/\rho}$, then with probability at least $1 - \iota$, the output $y$ satisfies that*

$$F(y) - F(x_0) \leq -\Phi,$$

*where $\Phi = \Omega(\frac{\gamma^{3/2}}{\sqrt{\rho} \log^3(dMZb/\rho\gamma\iota)})$.*

We have the following guarantee on $g_t$:

**Lemma 3.5.** *With probability at least $1 - \iota/2$, for all $t \in [\Gamma]$, we have*
$$\|g_{t-1} - \nabla F(x_{t-1})\|_2 \leq 2\gamma + (\varkappa + \rho\Xi)\Xi.$$

Let $\varrho_t = \nabla F(x_t) - g_t$ denote the difference between the true gradient and our estimation. We use the following standard technical claims to connect the distance of the trajectory and the function value change.

**Claim 3.6** (Lemma 10 in Wang et al. (2019a))**.** *We have*
$$F(x_{t+1}) - F(x_t) \leq -\frac{\eta}{4}\|\nabla F(x_t)\|_2^2 + 5\eta\|\varrho_t\|_2^2,$$

**Claim 3.7** (Lemma 11 in Wang et al. (2019a))**.** *For any $t \in [\Gamma]$, one has*

$$\|x_t - x_0\|_2^2 \leq 8\eta\Gamma(F(x_0) - F(x_t)) + 50\eta^2 t \sum_{i=1}^{t} \|\varrho_i\|_2^2.$$

In Algorithm 2, we halt the algorithm whenever we find a point $x_t$ such that $\|x_t - x_0\|_2 \geq \Xi$. We use the following lemma to demonstrate that a large distance at $x_t$ means a large function value decrease at $x_t$, that is, it escapes from the saddle point successfully:

**Lemma 3.8.** *Suppose Algorithm 2 halts in advance when condition $\|x_t - x_0\| \geq \Xi$ is satisfied, then we have*
$$F(x_t) - F(x_0) \leq -\Phi.$$

Similar to previous works Jin et al. (2017); Wang et al. (2019a), we use a coupling argument to prove that we can escape the saddle point with high probability. Fix the sequence of Gaussian noise $(\zeta_1, \cdots, \zeta_\Gamma)$ and ensure $\|\zeta_t\|_2 \leq \gamma$ for all $t \in [\Gamma]$.

Let $y(x)$ denote the output $y$ conditional on the first iterate $x_1 = x$. Define the set of stuck region around $x_0 - \eta g$:

$$\mathcal{X}(x_0) = \{x \mid x \in B(x_0 - \eta g, r), \text{ and } \|y(x) - x_0\|_2 \leq \Xi\}, \tag{1}$$

where $r = O(\eta\gamma)$. Suppose $x_0$ is the saddle point, and let $\mathbf{e}_1$ be the minimum eigenvector of $H$. We have the following Lemma:

**Lemma 3.9.** *Suppose $\varkappa \leq \sqrt{\rho\gamma}/2$. For any two points $w, u \in B(x_0, r)$, if $w - u = \mu r \mathbf{e}_1$ with $\mu \geq \iota^2/16\sqrt{d}$, then at least one of $w, u$ is not in $\mathcal{X}(x_0)$.*

With these Lemmas, we can complete the proof of Theorem 3.4. In particular, Theorem 3.4 suggests that, if we meet a saddle point, then after the following $\tilde{O}(1/\sqrt{\gamma})$ steps, the function value will decrease by at least $\tilde{\Omega}(\gamma^{3/2})$. This means the function value decreases by $\tilde{\Omega}(\gamma^2)$ on average for each step.

## 3.2 MAIN ALGORITHM

Algorithm 3 follows the SpiderBoost framework. We primarily focus on gradient oracles and estimators for simplicity, as the Hessian case follows the same principle. Moreover, we allow a larger error tolerance for a Hessian estimate $H_t$, approximately $\sqrt{\rho\gamma}$, compared to the gradient, which is $\gamma$. We discuss some key variables and parameters in it.

We either query $\mathcal{O}_1$ to estimate the gradient directly or query $\mathcal{O}_2$ to estimate the gradient difference between consecutive points. The term $\mathrm{drift}$ controls the estimated error: when $\mathrm{drift}_t$ is small, $\Delta_t$ remains a reliable estimator; when $\mathrm{drift}_t$ exceeds the threshold determined by the parameter $\kappa$, we obtain a fresh estimate from $\mathcal{O}_1$.

The term $\mathrm{frozen}$ serves a technical role in the application of Theorem 3.4; specifically, when a potential saddle point is detected, we invoke the subprocedure described in Algorithm 2.

The variable $\mathrm{count}$ tracks the number of saddle point escapes since the last fresh gradient estimate $\Delta_t$ and Hessian estimate $H_t$. Once $\mathrm{count}$ exceeds the threshold $\tau$, we force a fresh query to $\mathcal{O}_1$ and $\mathcal{O}_3$ to ensure privacy, following the Gaussian Mechanism.

We have the following guarantee of Algorithm 3.

---

**Algorithm 3** Stochastic Spider with Escaping Saddle Point SubProcedure

---
1: **Input:** Dataset $\mathcal{D}$, privacy parameters $\varepsilon, \delta$, parameters of objective function $B, M, G, \rho$, parameter $\kappa$, failure probability $\omega$, batch size parameter $b$, noise parameters $\sigma_{\text{tree}}, \{\sigma_t\}$
2: set $\text{drift}_0 = \kappa, \text{frozen}_{-1} = 1, \Delta_{-1} = 0, \mathcal{D}_r \leftarrow \mathcal{D}, t = 0, \text{EscapeFlag} = \textbf{False}$
3: **while** $t < T$ and the number of unused functions is larger than $b$ **do**
    // Estimate gradient and Hessian with oracle (Algorithm 4)
4:    **if** $\text{drift}_t \geq \kappa$ or $\text{count} \geq \tau$ **then**
5:       $\nabla_t = \mathcal{O}_1(x_t), H_t = \mathcal{O}_3(x_t),$
6:       $\text{drift}_t = 0, \text{frozen}_t = \text{frozen}_{t-1} - 1, \text{count} = 0$
7:    **else**
8:       $\Delta_t = \mathcal{O}_2(x_t, x_{t-1}), \nabla_t = \nabla_{t-1} + \Delta_t, \widetilde{\nabla}_t = \nabla_t + \text{TREE}(t)$
9:       $\Delta_t^H = \mathcal{O}_4(x_t, x_{t-1}), H_t = H_{t-1} + \Delta_t^H$
10:    **end if**
    // Escape Saddle Point if $\text{EscapeFlag} = \textbf{True}$
11:    **if** $\|\widetilde{\nabla}_t\| \leq \gamma \log^3(BMd/\rho\omega) \bigwedge \text{frozen}_{t-1} \leq 0$ **then**
12:       Set $\text{frozen}_t = \Gamma, \text{EscapeFlag} = \textbf{True}, g = \widetilde{\nabla}_t, H = H_t, x_{\text{anchor}} = x_t, \text{count} = \text{count} + 1$
13:    **end if**
14:    **if** $\text{EscapeFlag} == \textbf{True}$ **then**
15:       $g_t = g + H(x_t - x_{\text{anchor}}) + \xi_t$, where $\xi_t \sim \mathcal{N}(0, \sigma_t^2 I_d)$
16:    **else**
17:       $g_t = \widetilde{\nabla}_t$                                   ▷ Normal gradient descent
18:    **end if**
19:    $x_{t+1} = x_t - \eta g_t, \text{drift}_{t+1} = \text{drift}_t + \|g_t\|_2, \text{frozen}_{t+1} = \text{frozen}_t - 1,$
20:    **if** $\|x_{t+1} - x_{\text{anchor}}\| \geq \Xi$ or $\text{frozen}_{t+1} \leq 0$ **then**
21:       $\text{EscapeFlag} = \textbf{False}$
22:    **end if**
23:    $t = t + 1$
24: **end while**
25: **Return:** $\{x_1, \cdots, x_t\}$

---

**Proposition 3.10.** *Under Assumption 3.1 and with oracles such that $\|\widetilde{\nabla}_t - \nabla F(x_t)\| \leq \gamma$ and $\|H_t - \nabla^2 F(x_t)\|_2 \leq \sqrt{\rho\gamma}/2$ for any $t \in [T]$. When $\sigma_{\text{tree}} = \sigma_t = \frac{\gamma}{\sqrt{d \log(T/\iota)}}, \Gamma = \tilde{O}(M/\sqrt{\rho\gamma})$, $\varkappa\Xi \leq \gamma, \eta = 1/M$ and $\Xi = \sqrt{\gamma/\rho}$, setting $T = \tilde{O}(B/\eta\gamma^2)$, and supposing it does not halt before completing all $T$ iterations, with probability at least $1 - T\iota$, at least one point in the output set $\{x_1, \cdots, x_T\}$ of Algorithm 3 is $\tilde{O}(\gamma)$-SOSP.*

The proof intuition of Proposition 3.10 is, if we do not find an $O(\gamma)$-SOSP, then on average, the function value will at least decrease by $\Omega(\eta/\gamma^2)$. As we know $F_{\mathcal{P}}(x_0) \leq F_{\mathcal{P}}^* + B$, hence $O(B/\eta\gamma^2)$ steps can ensure we find an $O(\gamma)$-SOSP. See the proof of Proposition 3.10 in the Appendix.

It suffices to build the private oracles with provable utility guarantees. We construct the gradient oracles below in Algorithm 4.

**Lemma 3.11** (Oracles with bounded error). *Under assumption 3.1, let $\iota > 0$ and use Algorithm 4 as instantiations of $\mathcal{O}_1$ and $\mathcal{O}_2$. If $\mathcal{D}$ is i.i.d. drawn from distribution $\mathcal{P}$, we have:*
*(1) for any $x_t$, we have $\mathbb{E}[\mathcal{O}_1(x_t)] = \nabla F(x_t)$ and*

$$\Pr[\|\mathcal{O}_1(x_t) - \nabla F(x_t)\| \geq \zeta_1] \leq \iota,$$

*where $\zeta_1 = O(G\sqrt{\log(d/\iota)/b})$.*
*(2) for any $x_t, x_{t-1}$, we have $\mathbb{E}[\mathcal{O}_2(x_t, x_{t-1})] = \nabla F(x_t) - \nabla F(x_{t-1})$ and*

$$\Pr[\|\mathcal{O}_2(x_t, x_{t-1}) - (\nabla F(x_t) - \nabla F(x_{t-1}))\| \geq \zeta_2] \leq \iota,$$

*where $\zeta_2 = O(M\|x_t - x_{t-1}\|\sqrt{\log(d/\iota)/b_t})$.*
*(3) for any $x_t$. we have $\mathbb{E}[\mathcal{O}_3(x_t)] = \nabla^2 F(x_t)$ and*

$$\Pr[\|\mathcal{O}_3(x_t) - \nabla^2 F(x_t)\| \geq \zeta_3] \leq \iota,$$

where $\zeta_3 = O(M\sqrt{\log(d/\iota)/b})$.

*(4)for any $x_t, x_{t-1}$, we have $\mathbb{E}[\mathcal{O}_4(x_t, x_{t-1})] = \nabla^2 F(x_t) - \nabla^2 F(x_{t-1})$ and*

$$\Pr[\|\mathcal{O}_4(x_t, x_{t-1}) - (\nabla F(x_t) - \nabla F(x_{t-1}))\| \geq \zeta_4] \leq \iota,$$

*where $\zeta_4 = O(\rho\|x_t - x_{t-1}\|\sqrt{\log(d/\iota)/b_t})$.*

From now on, we adopt Algorithm 4 as the gradient oracles for Line 5 and Line 9 respectively in Algorithm 3, and we set $\eta = 1/M$. We then bound the error between gradient estimator $\nabla_t$ and the true gradient $\nabla F(x_t)$ for Algorithm 3.

**Lemma 3.12.** *Suppose the dataset is drawn i.i.d. from the distribution $\mathcal{P}$. For any $1 \leq t \leq T$ and letting $\tau_t \leq t$ be the largest integer such that $\mathrm{drift}_{\tau_t}$ is set to be 0, with probability at least $1 - T\iota$, for some universal constant $C > 0$, we have*

$$\|\nabla_t - \nabla F(x_t)\|^2 \leq C(\frac{G^2}{b} + \sum_{i=\tau_t+1}^{t} M^2\|x_i - x_{i-1}\|^2/b_i)\log(Td/\iota), \tag{2}$$

$$\|H_t - \nabla^2 F(x_t)\|^2 \leq C(\frac{M^2}{b} + \sum_{i=\tau_t+1}^{t} \rho^2\|x_i - x_{i-1}\|^2/b_i)\log^2(Td/\iota). \tag{3}$$

Now, we consider the noise added from the tree mechanism and the noise added in the stochastic power method to make the algorithm private.

**Lemma 3.13.** *If we set $\sigma_{\mathrm{tree}} = (\frac{G}{b} + \max_t \frac{\|\widetilde{\nabla}_t\|}{b_t})\log(1/\delta)/\varepsilon$ in the tree mechanism (Algorithm 1), $\sigma_t = (\frac{M}{b} + \max_t \frac{\rho\|x_t - x_{t-1}\|}{b_t}) \cdot \frac{2\Xi\sqrt{\Gamma\tau\log(1/\delta)}}{\varepsilon}$ and use Algorithm 4 as oracles, then Algorithm 3 is $(\varepsilon, \delta)$-DP.*

---

**Algorithm 4** oracles

1: **gradient oracle** $\mathcal{O}_1$
2: **inputs:** $x_t$
3: draw a batch size of $b$ among unused functions
4: **return:** $\frac{1}{b}\sum_z \nabla f(x_t; z)$

---

1: **gradient difference oracle** $\mathcal{O}_2$
2: **inputs:** $x_t, x_{t-1}$
3: draw a batch size of $b_t$ among unused functions
4: **return:** $\frac{1}{b_t}\sum_z (\nabla f(x_t; z) - \nabla f(x_{t-1}; z))$

---

1: **Hessian oracle** $\mathcal{O}_3$
2: **inputs:** $x_t$
3: draw a batch size of $b$ among unused functions
4: **return:** $\frac{1}{b}\sum_z \nabla^2 f(x_t; z)$

---

1: **Hessian difference oracle** $\mathcal{O}_4$
2: **inputs:** $x_t, x_{t-1}$
3: draw a batch size of $b_t$ among unused functions
4: **return:** $\frac{1}{b_t}\sum_z (\nabla^2 f(x_t; z) - \nabla^2 f(x_{t-1}; z))$

---

With the noise added in mind, we get the high-probability error bound of gradient estimators $\widetilde{\nabla}_t$ and Hessian estimators $H_t$.

**Lemma 3.14.** *In Algorithm 3, setting fixed batch size $b = G\sqrt{d}/\varepsilon\alpha + G^2/\alpha^2 + \frac{M\sqrt{d}}{\rho\varepsilon} + \frac{M^2}{\rho\alpha}$, adaptive batch size $b_t = \max\{\frac{\|g_t\|\cdot\sqrt{d}}{\alpha\varepsilon}, \frac{\kappa\cdot\|g_t\|}{\alpha^2}, \frac{\rho\kappa\cdot\|g_t\|}{M^2\alpha}, 1\}$, $\Xi = \sqrt{\gamma/\rho}, \tau = \frac{\alpha^{3/2}}{M\sqrt{\rho}}, \Gamma = \tilde{O}(M/\sqrt{\rho\gamma})$, $\sigma_t = 2\log(1/\delta)\alpha/\sqrt{d}, \forall t$ and $\sigma_{\mathrm{tree}} = 2\log(1/\delta)\alpha/\sqrt{d}$ correspondingly according to Lemma 3.13, for each $t \in [T]$, we know Algorithm 3 is $(\varepsilon, \delta)$-DP, and with probability at least $1-\iota$, $\|g_t - \nabla F(x_t)\| \leq \gamma$, $\|H_t - \nabla^2 F(x_t)\|_2 \leq \sqrt{\rho\gamma}/2$, where $\gamma = \tilde{O}(\alpha)$.*

We need to show that we can find an $\gamma$-SOSP before we use up all the functions. We need the following technical Lemma:

**Lemma 3.15.** *Consider the implementation of Algorithm 3. Suppose the size of dataset $\mathcal{D}$ can be arbitrarily large with functions drawn i.i.d. from $\mathcal{P}$, and we run the algorithm until finding an $\tilde{O}(\gamma)$-SOSP, then with probability at least $1 - T\iota$, the total number of functions we will use is bounded by*

$$\tilde{O}\Big(\frac{bBM}{\kappa\gamma} + BM(\frac{\sqrt{d}}{\gamma^2\varepsilon} + \frac{\kappa}{\gamma^3} + \frac{\rho\kappa}{M^2\gamma^2} + \frac{1}{\gamma^2})\Big).$$

Given the dataset size requirement, we can get the final bound on finding SOSP.

**Lemma 3.16.** *With $\mathcal{D}$ of size $n$ drawn i.i.d. from $\mathcal{P}$, setting $\kappa = \max\{\frac{\alpha\sqrt{d}}{\varepsilon}, (BGM)^{1/3}\}$,*

$$\alpha = O\Big(((BGM)^{1/3} + \sqrt{BM})(\frac{\sqrt{d}}{n\varepsilon})^{1/2} + \frac{B^{\frac{2}{9}}M^{\frac{2}{9}}G^{\frac{5}{9}} + B^{\frac{4}{9}}M^{\frac{4}{9}}G^{\frac{1}{9}}}{n^{1/3}}\Big)$$
$$+ O\Big(\frac{(MB)^{1/2}}{\sqrt{n}} + \frac{\sqrt{BM^2}}{\rho\sqrt{n}} + \frac{B^{1/3}M^{4/3}}{G^{1/6}\sqrt{\rho n}} + \frac{B^{2/3}G^{1/6}\sqrt{\rho}}{M^{1/3}\sqrt{n}} + \frac{B\rho\sqrt{d}}{Mn\varepsilon}\Big),$$

*and other parameters as in Lemma 3.14, with probability at least $1 - \iota$, at least one of the outputs of Algorithm 3 is $\gamma$-SOSP, with $\gamma = \tilde{O}(\alpha)$.*

Combining Lemma 3.13 and Lemma 3.16, we have the following main result for finding SOSP privately:

**Theorem 3.17.** *Given $\delta > 0, \varepsilon \leq O(1)$, with gradient oracles in Algorithm 4, setting $b = G\sqrt{d}/\varepsilon\alpha + G^2/\alpha^2 + \frac{M\sqrt{d}}{\rho\varepsilon} + \frac{M^2}{\rho\alpha}, b_t = \max\{\frac{\|g_t\| \cdot \sqrt{d}}{\alpha\varepsilon}, \frac{\kappa \cdot \|g_t\|}{\alpha^2}, \frac{\rho\kappa \cdot \|g_t\|}{M^2\alpha}, 1\}, \kappa = \max\{\frac{\alpha\sqrt{d}}{\varepsilon}, (BGM)^{1/3}\}, \Xi = \sqrt{\gamma/\rho}, \tau = \frac{\alpha^{3/2}}{M\sqrt{\rho}}, \Gamma = \tilde{O}(M/\sqrt{\rho\gamma})$, and $\sigma_{\text{tree}} = \sigma_t = \alpha/\sqrt{d}, \forall t$, Algorithm 3 is $(\varepsilon, \delta)$-DP, and if the dataset is i.i.d. drawn from the underlying distribution $\mathcal{P}$, at least one of its outputed points is $\tilde{O}(\alpha)$-SOSP, where*

$$\alpha = O\Big(((BGM)^{1/3} + \sqrt{BM})(\frac{\sqrt{d}}{n\varepsilon})^{1/2} + \frac{B^{\frac{2}{9}}M^{\frac{2}{9}}G^{\frac{5}{9}} + B^{\frac{4}{9}}M^{\frac{4}{9}}G^{\frac{1}{9}}}{n^{1/3}}\Big)$$
$$+ O\Big(\frac{(MB)^{1/2}}{\sqrt{n}} + \frac{\sqrt{BM^2}}{\rho\sqrt{n}} + \frac{B^{1/3}M^{4/3}}{G^{1/6}\sqrt{\rho n}} + \frac{B^{2/3}G^{1/6}\sqrt{\rho}}{M^{1/3}\sqrt{n}} + \frac{B\rho\sqrt{d}}{Mn\varepsilon}\Big),$$

*Remark* 3.18. If we treat the parameters $B, G, M, \rho$ as constants $O(1)$, then we get $\alpha = \tilde{O}((\frac{\sqrt{d}}{n\varepsilon})^{1/2} + \frac{1}{n^{1/3}})$ as claimed before in the abstract and introduction.

If we make further assumptions, like assuming the functions are defined over a constraint domain $\mathcal{X} \subset \mathbb{R}^d$ of diameter $R$ and we allow exponential running time, we can get some other standard bounds that can be better than Theorem 3.17 in some regimes. See Appendix C for more discussions.

## 4 DISCUSSION

We combine the concepts of adaptive batch sizes and the tree mechanism to improve the previous best results for privately finding SOSP. Our approach achieves the same bound as the state-of-the-art method for finding FOSP, suggesting that privately finding an SOSP may incur no additional cost.

Several interesting questions remain. First, what is the tight bound for privately finding FOSP and SOSP? Second, can the adaptive batch size technique be applied in other settings? Could it offer additional advantages, such as reducing runtime in practice? Finally, while we can obtain a generalization error bound of $\sqrt{d/n}$ using concentration inequalities and the union bound, can we achieve a better generalization error bound for the non-convex optimization?

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

## A OMITTED PROOF OF SUBSECTION 3.1

### A.1 PROOF OF LEMMA 3.5

*Proof.* By the concentration of the Gaussian, we know with probability at least $1 - \iota/2$, $\|\zeta_t\|_2 \leq \gamma$ for all $t \in [\Gamma]$. The following proof is conditional on the event that $\|\zeta_t\|_2 \leq \gamma$ for all $t$.

By the Assumption 3.1, for each $t \in [\Gamma]$, we know

$$
\begin{aligned}
\|g_{t-1} - \nabla F(x_{t-1})\|_2 &= \|g + H(x_{t-1} - x_0) + \zeta_{t-1} - \nabla F(x_{t-1})\|_2 \\
&\leq \|g + H(x_{t-1} - x_0) - \nabla F(x_{t-1})\|_2 + \|\zeta_{t-1}\|_2 \\
&\leq \|g + H(x_{t-1} - x_0) - \nabla F(x_{t-1})\|_2 + \gamma \\
&= \|g - \nabla F(x_0) + H(x_{t-1} - x_0) - (\nabla F(x_{t-1}) - \nabla F(x_0)\|_2 + \gamma \\
&\leq 2\gamma + \|(H - \nabla^2 F(z))(x_{t-1} - x_0)\|_2,
\end{aligned}
$$

where $z$ is a point in the section between $x_{t-1}$ and $x_0$. Note that $\|x_{t-1} - x_0\|_2 < \Xi$ by the algorithm design, hence we know $\|H - \nabla^2 F(z)\|_2 \leq \varkappa + \rho\Xi$. Hence, we have

$$\|g_{t-1} - \nabla F(x_{t-1})\|_2 \leq 2\gamma + (\varkappa + \rho\Xi)\Xi.$$

This completes the proof.

$\square$

### A.2 PROOF OF LEMMA 3.8

*Proof.* By Claim 3.6, we have

$$F(x_t) - F(x_0) \leq -\frac{\eta}{4} \sum_{i=1}^{t} \|\nabla F(x_i)\|_2^2 + 5\eta \sum_i^t \|\varrho_i\|_2^2. \tag{4}$$

Note that $x_t - x_0 = \eta(\sum_{i=1}^t \nabla F(x_i) - \varrho_i)$, which means $\eta\|(\sum_{i=1}^t \nabla F(x_i) - \varrho_i)\|_2 \geq \Xi$. By Lemma 3.5 and the preconditions, we know

$$\|\varrho_i\| \leq 3\gamma.$$

Hence we know

$$\eta\| \sum_{i=1}^{t} \nabla F(x_i)\| \geq \Xi - 3\gamma\Gamma \geq \sqrt{\gamma/2\rho}.$$

Then by Equation 4, we know

$$
\begin{aligned}
F(x_t) - F(x_0) &\leq -\frac{\eta}{4\Gamma}(\sum_{i=1}^{t}\|\nabla F(x_i)\|)^2 + 45\eta\Gamma\gamma^2 \\
&\leq -\frac{\eta\gamma}{8\Gamma\rho} + 45\eta\Gamma\gamma^2 \\
&\leq -\Phi.
\end{aligned}
$$

$\square$

### A.3    PROOF OF LEMMA 3.9

*Proof.* For proof purposes, let $\{x_t\}_{t\in[\Gamma]}$ and $\{x'_t\}_{t\in[\Gamma]}$ be the two trajectories with $x_1 = w$ and $x'_1 = u$.

It suffices to show that

$$
\max\{\|x_\Gamma - x_0\|_2, \|x'_\Gamma - x_0\|_2\} \geq \Xi. \tag{5}
$$

Let $z_t = x_t - x'_t$ be the difference. Hence we have

$$
\begin{aligned}
z_{t+1} &= z_t - \eta[g_{t-1} - g'_{t-1}] \\
&= z_t - \eta H(x_t - x'_t) \\
&= (I - \eta H)z_t.
\end{aligned}
$$

Recall that $z_1 = \mu r \mathbf{e}_1$, $\lambda_{\min}(\nabla^2 F(x_0)) \leq -\sqrt{\rho\alpha}$ and $\lambda_{\min}(H) \leq \sqrt{\rho\alpha}/2$. Then we know that

$$
\|z_t\| \geq (1 + \eta\sqrt{\rho\gamma}/2)^t \mu r,
$$

which means

$$
\|z_\Gamma\|_2 \geq 2\Xi
$$

by our choices of parameters. This establishes Equation (5) and completes the proof.

$\square$

### A.4    PROOF OF PROPOSITION 3.10

*Proof of Proposition 3.10.* By Lemma 3.2 and the precondition that $\|\widetilde{\nabla}_t - \nabla F(x_t)\| \leq \gamma$, we know that, if $\|\nabla F(x_t)\| \geq 4\gamma$, then $F(x_{t+1}) - F(x_t) \leq -\eta\|\widetilde{\nabla}\|^2/16$. Otherwise, $\|\nabla F(x_t)\| < 4\gamma$. If $\|\nabla F(x_t)\| < 4\gamma$ but $x_t$ is a saddle point, then by Theorem 3.4, we know with probability at least $1 - \iota$,

$$
F(x_{t+\Gamma}) - F(x_t) \leq -\tilde{\Omega}(\gamma^{3/2}/\sqrt{\rho}),
$$

where $\Gamma = \tilde{O}(M/\sqrt{\rho\gamma})$. Then if none of the points in $\{x_i\}_{i\in[T]}$ is an $\tilde{O}(\gamma)$-SOSP, then we know $F(x_T) - F(x_0) < -B$, which is contradictory to Assumption 3.1. Hence at least one point in $\{x_i\}_{i\in[T]}$ should be an $\tilde{O}(\gamma)$-SOSP, and hence complete the proof.

$\square$

### A.5    PROOF OF THEOREM 3.4

Now we complete the proof of Theorem 3.4.

*Proof of Theorem 3.4.* By Lemma 3.8 and the definition of $\mathcal{X}(x_0)$ (see Equation (1)), we have

$$
\Pr[F(y) - F(x_0) \leq -\Phi] \geq \Pr\left[x_1 \notin \mathcal{X}(x_0) \mid \|\zeta_t\|_2 \leq \gamma, \forall t\right] + \iota/2.
$$

It suffices to upper bound $\Pr\left[x_1 \in \mathcal{X}(x_0) \mid \|\zeta_t\|_2 \le \gamma, \forall t\right]$. Let $\mu_0 = \iota^2/4\sqrt{d}$. Recall the Gaussian we add is $\sigma_0 = \frac{\gamma}{\sqrt{d \log(T/\iota)}}$. Suppose we add two independent Gaussians $\zeta_0, \zeta_0'$ and get two independent first iterates $x_1$ and $x_1'$. By the property of Gaussians, we have

$$\Pr[|\mathbf{e}_1, \zeta_0 - \zeta_0'| \le \mu\gamma] = 2\mathrm{erf}(\mu\gamma/\sigma_0) \le \iota^2/16.$$

Then

$$\Pr\left[|\mathbf{e}_1, \zeta_0 - \zeta_0'| \le \mu\gamma \mid \|\zeta_0\| \le \gamma, \|\zeta_0'\| \le \gamma\right] \le \frac{\iota^2/16}{1-\iota} \le \iota^2/4.$$

By Lemma 3.9 and independence between $x_1, x_1'$, we have

$$\begin{aligned}
\Pr\left[x_1 \in \mathcal{X}(x_0) \mid \|\zeta_t\|_2 \le \gamma, \forall t\right] &\le \sqrt{\Pr[x_1, x_1' \in \mathcal{X}(x_0) \mid \|\zeta_0\| \le \gamma, \|\zeta_0'\| \le \gamma]} \\
&\le \sqrt{\Pr\left[|\mathbf{e}_1, \zeta_0 - \zeta_0'| \le \mu\gamma \mid \|\zeta_0\| \le \gamma, \|\zeta_0'\| \le \gamma\right]} \\
&\le \iota/2.
\end{aligned}$$

Hence, we show that

$$\Pr[F(y) - F(x_0) \le -\Phi] \ge 1 - \iota.$$

$\square$

## B    PROOF OF SUBSECTION 3.2

### B.1    PROOF OF LEMMA 3.11

*Proof.* We only prove the first two items, as (3) and (4) follow from similar arguments. For each data $z \sim \mathcal{P}$, we know

$$\mathbb{E}\,\nabla f(x_t; z) - \nabla F(x_t) = 0, \quad \|\nabla f(x_t; z) - \nabla F(x_t)\| \le 2G.$$

Then the conclusion (1) follows from Lemma 2.4.

Similarly, for each data $z \sim \mathcal{P}$, we know

$$\mathbb{E}(\nabla f(x_t; z) - \nabla f(x_{t-1}; z)) - (\nabla F(x_t; z) - \nabla F(x_{t-1}; z)) = 0,$$
$$\|(\nabla f(x_t; z) - \nabla f(x_{t-1}; z)) - (\nabla F(x_t; z) - \nabla F(x_{t-1}; z))\| \le 2M\|x_t - x_{t-1}\|.$$

The statement (2) also follows from Lemma 2.4. $\square$

### B.2    PROOF OF LEMMA 3.12

*Proof.* We consider the case when $t = \tau_t$ first, i.e., we query $\mathcal{O}_1$ to get $\nabla_t$. Then Equation (2) follows from Lemma 3.11.

When $t > \tau_t$, then for each $i$ such that $\tau_t < i \le t$, we know conditional on $\nabla_{i-1}$, we have

$$\mathbb{E}[\Delta_i \mid \nabla_{i-1}] = \nabla F(x_i) - \nabla F(x_{i-1}).$$

That is $\Delta_i - (\nabla F(x_i) - \nabla F(x_{i-1}))$ is zero-mean and $\mathrm{nSG}(M\|x_i - x_{i-1}\|\sqrt{\log(dT/\iota)}/\sqrt{b_i})$ by applying Lemma 3.11. Then Equation (2) follows from applying Lemma 2.4.

The case for Hessian estimation involves some truncation. By Lemma 3.11, we know with probability at least $1 - T\iota$, we have

$$\Pr[\|H_{\tau_t} - \nabla^2 F(x_{\tau_t})\|_2 \ge \zeta_3] \le \iota/T,$$

with $\zeta_3 = O(M\sqrt{\log(Td/\iota)/b})$. The similar concentration holds for $\Delta_t^H$. We truncate the distribution of $H_{\tau_t}$ around $\nabla^2 F(x_{\tau_t})$, that is $\overline{H}_{\tau_t} = H_{\tau_t} \cdot \mathbf{1}\left(\|H_{\tau_t} - \nabla^2 F(x_{\tau_t})\| \le \zeta_3\right)$. It is straightforward to see that

$$\|\mathbb{E}\,\overline{H}_{\tau_t} - \nabla^2 F(x_{\tau_t})\| \le \int_{\zeta_3}^{\infty} \Pr[\|H_{\tau_t} - \nabla^2 F(x_{\tau_t})\| \ge \zeta]\mathrm{d}\zeta \le \zeta_3/T.$$

Truncate $\Delta_t^H$ in a similar way. Then by Lemma 3.11 we can show that

$$\Pr[\|\overline{H}_{\tau_t} + \sum_{i=\tau_t+1}^{t} \overline{\Delta}_i^H - \mathbb{E}[\overline{H}_{\tau_t} + \sum_{i=\tau_t+1}^{t} \overline{\Delta}_i^H]\|^2 \geq \frac{C}{2}(\frac{M^2}{b} + \sum_{i=\tau_t+1}^{t} \rho^2\|x_i - x_{i-1}\|^2/b_i)\log^2(Td/\iota)] \leq \iota.$$

(6)

Note that

$$\|\mathbb{E}[\overline{H}_{\tau_t} + \sum_{i=\tau_t+1}^{t} \overline{\Delta}_i^H] - \nabla^2 F(x_t)\| = \|\mathbb{E}[\overline{H}_{\tau_t} + \sum_{i=\tau_t+1}^{t} \overline{\Delta}_i^H] - \mathbb{E}[H_{\tau_t} + \sum_{i=\tau_t+1}^{t} \Delta_i^H]\| \quad (7)$$

$$\leq \frac{C\log(Td/\iota)}{2T}(\frac{M}{\sqrt{b}} + \sum_{i=\tau_t+1}^{t} \rho\|x_i - x_{i-1}\|/\sqrt{b_i}).$$

By union bound, we have

$$\Pr[\|\overline{H}_{\tau_t} + \sum_{i=\tau_t+1}^{t} \overline{\Delta}_i^H \neq H_t] \leq T\iota/2. \quad (8)$$

Equations (6),(7) and (8) complete the proof. $\qquad \square$

## B.3 PROOF OF LEMMA 3.13

*Proof.* We use the tree mechanism to privatize the gradients if no potential saddle point is met, and we use the Gaussian mechanism during escaping from the saddle point.

We first show the indistinguishability of the gradients. It suffices to consider the sensitivity of the gradient oracles.

Consider the sensitivity of $\mathcal{O}_1$ first. Let $\mathcal{O}(x_t)'$ denote the output with the neighboring dataset. Then it is obvious that

$$\|\mathcal{O}_1(x_t) - \mathcal{O}_1(x_t)'\| \leq \frac{G}{b}.$$

As for the sensitivity of $\mathcal{O}_2$, we have

$$\|\mathcal{O}_2(x_t, x_{t-1}) - \mathcal{O}_2(x_t, x_{t-1})'\| \leq \frac{M\|x_t - x_{t-1}\|}{b_t} = \frac{\|\widetilde{\nabla}_t\|}{b_t}.$$

The privacy guarantee of gradients follows from the tree mechanism (Theorem 2.2), which means $\{\widetilde{\nabla}_t\}_t \approx_{\varepsilon/2,\delta/2} \{\widetilde{\nabla}'_t\}_t$.

As for the Gaussian mechanism, note that for neighboring datasets with Hessian estimates $H_t$ and $H_t'$ respectively, we know that

$$\|H_t - H_t'\|_2 \leq \frac{M}{b} + \max_t \frac{\rho\|x_t - x_{t-1}\|}{b_t}.$$

Note that we force a fresh estimate from $\mathcal{O}_3$ after escaping the saddle points for $\tau$ times, and in each time, we ensure that $\|x_t - x_0\| \leq \Xi$ and $H_t$ are used at most $\Gamma$ steps, which means the difference is

$$\|(H_t - H_t')(x_t - x_0)\|_2 \leq \|H_t - H_t'\| \cdot \|x_t - x_0\| \leq (\frac{M}{b} + \max_t \frac{\rho\|x_t - x_{t-1}\|}{b_t}) \cdot \Xi.$$

Then the property of Gaussian Mechanism and composition finishes the privacy guarantee proof. $\qquad \square$

## B.4 PROOF OF LEMMA 3.14

*Proof.* By our setting of parameters, we know $\Xi\sqrt{\Gamma\tau} \leq \alpha/\rho$ and hence

$$(\frac{G}{b} + \max_t \frac{\|g_t\|}{b_t}) \leq 2\varepsilon\alpha/\sqrt{d},$$

$$(\frac{M}{b} + \max_t \frac{\rho\|x_t - x_{t-1}\|}{b_t}) \cdot (2\Xi\sqrt{\Gamma\tau}) \leq 2\varepsilon\alpha/\sqrt{d},$$

Then our choice of $\sigma_{\text{tree}}$ and $\sigma_t$ ensures the privacy guarantee by Lemma 3.13.

For any $t \in [T]$, if it is in the process of escaping from the saddle point, and $g_t = g + H(x_t - x_{\text{anchor}}) + \xi_t$, then Lemma 3.5 and our parameter settings ensure the accuracy on $g_t$. Consider the other case when $g_t = \widetilde{\nabla}_t$, we have

$$\|\widetilde{\nabla}_t - \nabla F(x_t)\| \le \underbrace{\|\widetilde{\nabla}_t - \nabla_t\|}_{(1)} + \underbrace{\|\nabla_t - \nabla F(x_t)\|}_{(2)}.$$

By Theorem 2.2, we know

$$(1) \le \max_t \|\text{TREE}(t)\| \le \sigma \sqrt{d \log(T)} \le \sigma \sqrt{d \log n} \lesssim \alpha \sqrt{\log n} \le \tilde{O}(\gamma).$$

By Lemma 3.12 and our parameter settings, we have

$$\begin{aligned}
\|\nabla_t - \nabla F(x_t)\|^2 &\lesssim (\alpha^2 + \sum_{i=\tau_t+1}^{t} \|g_t\|^2/b_t) \log(nd/\iota) \\
&\lesssim (\alpha^2 + \sum_{i=\tau_t+1}^{t} \|g_t\| \cdot \min\{\frac{\alpha\varepsilon}{\sqrt{d}}, \alpha^2/\kappa\}) \log(nd/\iota) \\
&\le (\alpha^2 + \kappa \cdot \min\{\frac{\alpha\varepsilon}{\sqrt{d}}, \alpha^2/\kappa\}) \log(nd/\iota) \\
&\lesssim \alpha^2 \log(nd/\iota).
\end{aligned}$$

Hence we conclude that $(2) \lesssim \alpha\sqrt{\log(nd/\iota)}$.

Similarly, by Lemma 3.12, we can conclude that

$$\begin{aligned}
\|H_t - \nabla^2 F(x_t)\|^2 &\lesssim (\rho\alpha + \sum_{i=\tau_t+1} \rho^2 \|g_t\|^2/M^2 b_t) \log^2(nd/\iota) \\
&\lesssim (\rho\alpha + \sum_{i=\tau_t+1}^{t} \rho^2 \|g_t\| \cdot (\alpha/\rho\kappa)) \log^2(nd/\iota) \\
&\lesssim \rho\alpha \log^2(nd/\iota),
\end{aligned}$$

which completes the proof. $\qquad\square$

## B.5 PROOF OF LEMMA 3.15

*Proof.* By Proposition 3.10, setting $T = \tilde{O}(B/\eta\gamma^2)$ suffices to find an $\tilde{O}(\gamma)$-SOSP. Let $\{x_1, \cdots, x_t\}$ be the outputs of the algorithms, where $t \le T$ denotes the step we halt the algorithm. We first show

$$\sum_{i=1}^{t} \|g_i\|_2 \lesssim \tilde{O}(BM/\gamma). \tag{9}$$

Denote the set $S := \{i \in [t] : \|g_i\| \le \gamma\}$. As $|S| \le T = \tilde{O}(B/\eta\gamma^2)$, we know $\sum_{i\in S} \|g_i\| \le \tilde{O}(B/\eta\gamma)$.

Now consider the set $S^c := [t] \setminus S$ denoting the index of steps when the norm of the gradient estimator is large. It suffices to bound $\sum_{i\in S^c} \|g_i\|_2$.

By Lemma 3.2, we know when $\|g_i\| \ge \gamma$, $F(x_{i+1}) - F(x_i) \le -\eta\|g_i\|^2/16$, and when $\|g_i\| \le \gamma$, $F(x_{i+1}) \le F(x_i) + \eta\gamma^2$. Given the bound on the function values, we know

$$\sum_{i\in S^c} \|g_i\|^2 \le \tilde{O}(B/\eta).$$

Hence

$$\sum_{i \in S^c} \|g_i\| \leq \frac{\sum_{i \in S^c} \|g_i\|^2}{\gamma} \leq \tilde{O}(\frac{B}{\eta\gamma}).$$

This completes the proof of Equation (9). Moreover, we know the total time that we will escape from the saddle point is at most $O(B/\Phi)$. Hence, The total number of functions we used for $\mathcal{O}_1$ and $\mathcal{O}_3$, is upper bounded by

$$b \cdot (\frac{\sum_{i \in [t]} \|g_i\|}{\kappa} + B/\Phi\tau) = \tilde{O}(bBM/\kappa\gamma).$$

The total number of functions we used for $\mathcal{O}_2$ is upper bounded as follows:

$$\sum_{i \in [t]} b_t \lesssim (\frac{\sqrt{d}}{\alpha\varepsilon} + \frac{\kappa}{\alpha^2} + \frac{\rho\kappa}{M^2\alpha}) \sum_{i \in [t]} \|g_i\| + T \leq BM \cdot \tilde{O}(\frac{\sqrt{d}}{\gamma^2\varepsilon} + \frac{\kappa}{\gamma^3} + \frac{\rho\kappa}{M^2\gamma^2} + \frac{1}{\gamma^2}).$$

This completes the proof. □

## B.6  PROOF OF LEMMA 3.16

*Proof.* By Lemma 3.15 and our parameter settings, we need

$$n \geq \tilde{\Omega}(\frac{bBM}{\kappa\gamma} + BM(\frac{\sqrt{d}}{\gamma^2\varepsilon} + \frac{\kappa}{\gamma^3} + \frac{\rho\kappa}{M^2\gamma^2} + \frac{1}{\gamma^2})).$$

First,

$$n \geq \tilde{\Omega}(\frac{bBM}{\kappa\gamma}) = \Theta(\frac{BGM\sqrt{d}}{\kappa\varepsilon\gamma^2} + \frac{G^2BM}{\kappa\gamma^3} + \frac{BM^2\sqrt{d}}{\kappa\rho\gamma\varepsilon} + \frac{BM^3}{\kappa\rho\gamma^2})$$

$$\Leftarrow \gamma \geq \tilde{O}(\frac{(BGM)^{1/3}d^{1/4}}{\sqrt{n\varepsilon}} + \frac{B^{\frac{2}{9}}M^{\frac{2}{9}}G^{\frac{5}{9}}}{n^{1/3}} + \frac{\sqrt{BM^2}}{\sqrt{\rho n}} + \frac{B^{1/3}M^{4/3}}{G^{1/6}\sqrt{\rho n}}).$$

Secondly,

$$n \geq \tilde{\Omega}(BM\sqrt{d}/\gamma^2\varepsilon) \Leftarrow \gamma \geq \tilde{O}(\sqrt{BM}(\frac{\sqrt{d}}{n\varepsilon})^{1/2}).$$

Thirdly,

$$n \geq \tilde{\Omega}(BM\kappa/\gamma^3) \Leftrightarrow n \geq \tilde{\Omega}(BM(\frac{\sqrt{d}}{\gamma^2\varepsilon}) + B^{4/3}M^{4/3}G^{1/3}/\gamma^3)$$

$$\Leftarrow \gamma \geq \tilde{O}(\sqrt{BM}(\frac{\sqrt{d}}{n\varepsilon})^{1/2} + \frac{B^{\frac{4}{9}}M^{\frac{4}{9}}G^{\frac{1}{9}}}{n^{1/3}}).$$

Fourthly,

$$n \geq \tilde{\Omega}(\frac{B\rho\kappa}{M\gamma^2}) \geq \tilde{\Omega}(\frac{B^{4/3}G^{1/3}\rho}{M^{2/3}\gamma^2} + \frac{B\rho\sqrt{d}}{M\gamma\varepsilon})$$

$$\Leftarrow \gamma \geq \tilde{O}(\frac{B^{2/3}G^{1/6}\sqrt{\rho}}{M^{1/3}\sqrt{n}} + \frac{B\rho\sqrt{d}}{Mn\varepsilon}).$$

Finally,

$$n \geq \tilde{\Omega}(MB/\gamma^2) \Leftarrow \gamma \geq \tilde{O}(\frac{(MB)^{1/2}}{\sqrt{n}}).$$

Combining these together, we get the claimed statement.

□

## C  OTHER RESULTS

The first result is combining the current result in finding the SOSP of the empirical function $F_{\mathcal{D}}(x) := \frac{1}{n} \sum_{\zeta \in \mathcal{D}} f(x; \zeta)$, and then apply the generalization error bound as follows:

**Theorem C.1.** *Suppose $\mathcal{D}$ is i.i.d. drawn from the underlying distribution $\mathcal{P}$ and under Assumption 3.1. Additionally assume $f(; \zeta) : \mathcal{X} \to \mathbb{R}$ for some constrained domain $\mathcal{X} \subset \mathbb{R}^d$ of diameter $R$. Then we know for any point $x \in \mathcal{X}$, with probability at least $1 - \iota$, we have*

$$\|\nabla F_{\mathcal{P}}(x) - \nabla F_{\mathcal{D}}(x)\| \leq \tilde{O}(\sqrt{d/n}), \|\nabla^2 F_{\mathcal{P}}(x) - \nabla^2 F_{\mathcal{D}}(x)\| \leq \tilde{O}(\sqrt{d/n}).$$

*Proof.* We construct a maximal packing $\mathcal{Y}$ of $O((R/r)^d)$ points for $\mathcal{X}$, such that for any $x \in \mathcal{X}$, there exists a point $y \in \mathcal{Y}$ such that $\|x - y\| \leq r$.

By Union bound, the Hoeffding inequality for norm-subGaussian (Lemma 2.4 and the Matrix Bernstein Inequality(Theorem 2.5), we know with probability at least $1 - \tau$, for all point $y \in \mathcal{Y}$, we have

$$\|\nabla F_{\mathcal{P}}(y) - \nabla F_{\mathcal{D}}(y)\| \leq \tilde{O}(L\sqrt{d \log(R/r)/n}), \|\nabla^2 F_{\mathcal{P}}(y) - \nabla^2 F_{\mathcal{D}}(y)\| \leq \tilde{O}(M\sqrt{d \log(R/r)/n}).$$
$$\tag{10}$$

Conditional on the above event Equation (10). Choosing $r \leq \min\{1, M/\rho\}\sqrt{d/n}$, then by the assumptions on Lipschitz and smoothness, we have for any $x \in \mathcal{X}$, there exists $y \in \mathcal{Y}$ such that $\|x - y\| \leq r$, and

$$\|\nabla F_{\mathcal{P}}(x) - \nabla F_{\mathcal{D}}(x)\| \leq \|\nabla F_{\mathcal{P}}(x) - \nabla F_{\mathcal{P}}(y)\| + \|\nabla F_{\mathcal{P}}(y) - \nabla F_{\mathcal{D}}(y)\| + \|\nabla F_{\mathcal{D}}(y) - \nabla F_{\mathcal{D}}(x)\|$$
$$\leq \tilde{O}(L\sqrt{d/n}).$$

Similarly, we can show

$$\|\nabla^2 F_{\mathcal{P}}(x) - \nabla^2 F_{\mathcal{D}}(x)\| \leq \tilde{O}((M + \rho r)\sqrt{d/n}) = \tilde{O}(M\sqrt{d/n}).$$

$\square$

The current SOTA of finding SOSP privately of $F_{\mathcal{D}}$ is from Ganesh et al. (2023), where they can find an $\tilde{O}((\sqrt{d}/n\varepsilon)^{2/3})$-SOSP. Combining the SOTA and Theorem C.1, we can find the $\alpha$-SOSP of $F_{\mathcal{P}}$ privately with

$$\alpha = \tilde{O}(\frac{\sqrt{d}}{n} + (\frac{\sqrt{d}}{n\varepsilon})^{2/3}).$$

If we allow exponential running time, as Lowy et al. (2024) suggests, we can find an initial point $x_0$ privately to minimize the empirical function and then use $x_0$ as a warm start to improve the final bound further.

