# OpenReview forum: "Adaptive Batch Size for Privately Finding Second-Order Stationary Points"
_ICLR.cc/2025/Conference — ICLR 2025 Spotlight_

### Official Review · Reviewer_cUiT · 2024-10-31

**Soundness:** 4
**Presentation:** 2
**Contribution:** 3
**Rating:** 8
**Confidence:** 4

**Summary:**

This paper presents a new algorithm for private stochastic optimization that builds on the SPIDER algorithm, incorporating a well-known differential privacy technique called the "tree mechanism" and an adaptive batch-size approach. The adaptive strategy adjusts the batch size based on a bias-variance criterion. The authors provide proofs showing that their algorithm achieves faster convergence rates to second-order stationary points than previous methods.

**Strengths:**

- The paper provides a theoretical improvement over the existing methods, particularly by achieving faster convergence to second-order stationary points under privacy constraints.
- The proofs are thorough and well-written, giving strong support to the theoretical claims.
- The paper gives sound reasons for developing the algorithm, which addresses both theoretical and practical needs in private stochastic optimization.

**Weaknesses:**

- The structure and flow could be improved, and feels a bit rushed. At times, the paper feels like a sequence of technical lemmas without enough overview or context, which can make it hard to follow.
- The paper does not include an empirical evaluation of the proposed method. Although the main focus is on theoretical improvement, a comparison with other methods in private stochastic optimization would be helpful to understand the practical performance of the algorithm.

**Questions:**

- The notation $\mathcal{O}_1$ and $\mathcal{O}_2$ for the gradient oracles could be confusing, as it resembles the common asymptotic notation $O$. An alternative notation might reduce ambiguity.
- How do the assumptions in this work compare to those in related papers? A comparison could add useful context.
- The current approach allows each data point to be seen only in one of the inner loop of the SPIDER algorithm. Could sample complexity be improved by sharing data across outer loops while tracking privacy using composition and subsampling amplification?

---

> ### Author Response · Authors · 2024-11-19
>
> Thank you for your valuable feedback and thoughtful comments. Below are our responses to your points and questions:
>
> 1. On Structure and Flow:
> We acknowledge that the structure and flow of the paper could be improved. We will add more discussions to provide intuition for the technical lemmas and ensure the paper feels more cohesive and easier to follow.
>
> 2. On Empirical Evaluations:
> While the primary focus of this work is on theoretical improvements, an empirical evaluation would provide additional insights. We believe that comparing our approach to existing algorithms in the literature could make for a valuable separate project.
>
> 3. On Notations:
> Thank you for pointing out the potential ambiguity in the notations $\mathcal{O}_1$ and $\mathcal{O}_2$​. We will consider revising the notation to reduce confusion.
>
> 4. Assumptions in the Work:
> Our assumptions (e.g., Lipschitz, smoothness, and Hessian Lipschitz conditions) are standard in the literature and align with those used in related works.
>
> 5. On Sample Complexity and Data Sharing (Question 3):
> This is an excellent question. Sharing data across the outer loops while maintaining unbiased gradients poses challenges, as reusing data points could introduce dependencies. As a result, the gradient estimates may no longer be unbiased. A naive generalization bound in a bounded domain is $O(\sqrt{d/n})$, as discussed in Appendix A.2. We currently do not have a satisfactory solution for addressing this issue but agree that it is an important direction for future investigation. This may improve the bound for finding SOSP of population function.

---

> > ### Comment · Reviewer_cUiT · 2024-11-22
> >
> > Thank you for your detailed response and the clarifications provided. I appreciate your plans to improve the structure and notation, as well as your thoughtful insights on sample complexity and data sharing. I have decided to raise my score from 6 to 8, as I believe this is a sound and solid paper with significant theoretical value.

---

> > > ### Author Response · Authors · 2024-11-22
> > >
> > > Thank you for your thoughtful feedback and for taking the time to consider our response carefully. We are truly grateful for your recognition of the theoretical value of our work and for raising your score. If you have any additional suggestions or concerns, please don’t hesitate to let us know.

---

### Official Review · Reviewer_pqBp · 2024-11-02

**Soundness:** 3
**Presentation:** 2
**Contribution:** 3
**Rating:** 6
**Confidence:** 3

**Summary:**

This paper improves the theoretical bounds for reaching a Second-Order Stationary Point (SOSP) under differential privacy constraints. By employing an adaptive batch size and replacing the independent Gaussian mechanism with a private tree mechanism, the paper effectively reduces the error in achieving SOSP. Detailed theoretical proofs are provided.

**Strengths:**

This paper addresses an important problem by improving the theoretical bound for achieving a second-order stationary point (SOSP) under differential privacy constraints, aligning it with the bound for a first-order stationary point (FOSP). Leveraging the tree mechanism in differentially private continuous observation, which has been shown to achieve asymptotically minimal error, the paper successfully reduces the error for SOSP.  I believe the idea is noval and the contribution is sufficient.

**Weaknesses:**

1. There is a line of work that further improves the theoretical bound of the tree mechanism, called matrix mechanism [1]. It improves the bound of the tree mechanism by a constant factor. When training a model, it generally works better than the tree mechanism [2]. I would like to hear some discussions about whether using the matrix mechanism can help improve the theoretical analysis.

2. I am confused by how you describe the adaptive batch size, the expression of $b_t$ in Lemma 3.8 depends on $\tilde{\Delta}_t$. Is  $\tilde{\Delta}_t$ private? If you need to change the batch size based on the dataset, there's a risk of leaking private information when the batch size is not protected. Could you explain how you make the batch size private?

3. As stated in point 2, the notations sometimes confuse me. If would be great if there is a table of notion provided, which would help the paper more readable.

[1] Fichtenberger, Hendrik, Monika Henzinger, and Jalaj Upadhyay. "Constant matters: Fine-grained error bound on differentially private continual observation." In International Conference on Machine Learning, pp. 10072-10092. PMLR, 2023.

[2] Choquette-Choo, Christopher A., Arun Ganesh, Ryan McKenna, H. Brendan McMahan, John Rush, Abhradeep Guha Thakurta, and Zheng Xu. "(Amplified) Banded Matrix Factorization: A unified approach to private training." Advances in Neural Information Processing Systems 36 (2024).

**Questions:**

A minor question, does the norm in the paper mean $l_2$ norm?

---

> ### Author Response · Authors · 2024-11-19
>
> Thank you for taking the time to review our work and for your detailed feedback. Below, we address your questions and comments:
>
> 1. Regarding the Matrix Mechanism:
> Within our algorithm framework, improvements over the binary tree mechanism primarily affect constants but will not affect the asymptotics. However, discussing the potential use of the matrix mechanism and its advantages would be valuable, and we will include such discussions in the revised version.
>
> 2. On the privacy of $\tilde{\Delta}_t$:
> The binary tree mechanism privates $\tilde{\Delta}_t$. Consequently, $b_t$ (specified in Line 369)  is also private due to the postprocessing property of differential privacy.
>
> 3. On Notation and Norms:
> As noted In line 164, we use $\|\cdot\|$ to represent the $\ell_2$ norm of a vector and the operator norm of a matrix. To improve clarity, we will expand the explanations of notations in the revised version and consider including a table of notation for better readability.

---

> > ### Comment · Reviewer_pqBp · 2024-11-21
> >
> > Thank you for the clarification!

---

> ### Author Response · Authors · 2024-11-21
>
> Thank you for your response and for acknowledging the clarifications. If you have any further questions or concerns, please don’t hesitate to reach out—we’d be happy to address them.
>
> Additionally, if our clarifications have adequately addressed your concerns, we kindly request you to consider revisiting the score, as it would greatly help us in the review process.
>
> Thank you again for your valuable time and feedback!

---

### Official Review · Reviewer_Yw5T · 2024-11-04

**Soundness:** 4
**Presentation:** 4
**Contribution:** 3
**Rating:** 8
**Confidence:** 3

**Summary:**

This paper studies privately finding second-order stationary points of stochastic optimization problems. This problem is a relevant to private machine learning. A related problem is privately finding first-order stationary points. First-order stationary points include saddle points and local optima, making them less valuable for machine learning compared to second-order stationary points, which exclude saddle points.

The paper gives a differentially private algorithm for finding second-order stationary points. Prior state-of-the-art had a rate higher than for first-order stationary points. This work achieves rate-parity between second-order and first-order stationary points under differential privacy.

The prior state-of-the-art algorithm involves infrequent gradient queries interspersed with many gradient-difference queries. The queries are answered by randomly subsampling the dataset by a fixed-size batch. Gaussian noise is introduced to ensure privacy. A drift variable tracks the total amount of l2-movement of the variable and a fresh gradient query is made when the drift crosses a threshold, which ensures that the gradient estimates do not deteriorate significantly.

This work improves upon the prior work in two ways. First, the Gaussian noise mechanism is replaced with a correlated noise mechanism that reduces the cumulative noise introduced by successive gradient-difference queries. Second, the batch-size for gradient-difference queries between two points is chosen adaptively in proportion to the l2-distance between the two points. Due to the underlying smoothness assumption, the sensitivity of the gradient-difference queries grows in proportion to this distance. Thus adaptive batch-sizes allows for more targeted noise levels and also tightens the utility-analysis.

**Strengths:**

The paper proposes a novel algorithm for an important problem in private machine learning. The algorithm improves meaningfully on the previous state-of-the-art in multiple clear ways.

The writing is generally clear and ideas are easy to follow.

**Weaknesses:**

My only concern is a lack of experimental results. It would be nice to see a small experiment comparing the algorithm to the prior state-of-the-art in order to give a sense of whether the new algorithm is practical to run or not as well as whether the theoretical improvements actually translate into substantive improvement in the quality of the SOSP. In particular, I would be interested to see how the runtime as well as the $\alpha$ value of your approach compares to prior approaches on benchmark problem instances of increasing size.

**Questions:**

I was a bit surpised that the private rate only improved from $\tilde{O}((\sqrt{d}/\epsilon n)^{3/7})$ to $\tilde{O}((\sqrt{d}/\epsilon n)^{1/2})$ after what seem at a high level like fairly substantial improvements to the algorithm. Is this bound known to be tight for FOSP?

> In our setting, $\mathcal{O}_1$ is more accurate but incurs higher computational or privacy costs.

This point is not entirely clear to me and it seems very foundational to the structure of the algorithm. Why shouldn't we just query $\mathcal{O}_1$ every time? I assume it is related to the sensitivity of the queries. It would be helpful for the authors to clarify the motivation for using both oracles by explicitly state the tradeoff between using $\mathcal{O}_1$ and $\mathcal{O}_2$ in terms of accuracy, privacy cost, and/or computational complexity.

> When the privacy parameter $\epsilon$ is sufficiently small, we observe that $\alpha_F \ll \alpha_S$.

I am not completely following this either. If $n$ and $d$ are fixed and we take $\epsilon \to 0$, then eventually we should have $(\sqrt{d}/\epsilon n)^{1/2} > (\sqrt{d}/\epsilon n)^{3/7}$. Could the authors explain the regime (in terms of relationships between $n$, $d$, and $\epsilon$) where the improvement is most significant and clarify whether there are any limitations of their approach as $\epsilon$ becomes very small?

---

> ### Author Response · Authors · 2024-11-19
>
> Thank you for your valuable feedback. While we aim to present a self-contained theoretical paper in this submission, empirical comparisons with existing algorithms in the literature would be an interesting direction for future work.
>
> Below, we address your questions:
>
> 1. For FOSP, the current lower bound we know is $\Omega(\sqrt(d)/n\epsilon)$ for privately finding FOSP of the empirical function, a result derived from private mean estimation (and applicable to SOSP as well). The non-private term $O(1/n^{1/3})$ is only known to be tight in the high-dimension case (as we discussed around Line 85). Closing these gaps for both FOSP and SOSP remains an open and intriguing research direction.
> 2. If we query oracle_1 every time, the tree mechanism offers no advantage, and the approach effectively reduces back to the DP-SGD combined with the Gaussian mechanism.  Even in non-private settings, variance-reduction-based algorithms generally outperform standard SGD (in terms of theoretical convergence rate).
> Lemma 3.6 provides some intuition for leveraging oracle_2: querying oracle_1 alone leads to an error proportional to $\sum_i ||x_i-x_{i-1}||$, whereas incorporating oracle_2 for variance reduction makes the error proportional to $\sqrt{\sum_i ||x_i-x_{i-1}||^2}$. This allows us to use a larger batch size each time we query oracle_1, significantly reducing the variance and error of the gradient estimators.
> 3. We are considering the regime where $\sqrt(d)/n\epsilon<< O(1)$, ensuring that the bounds remain non-trivial (as under the Lipschitz assumption, any point can be an O(1)-SOSP). In this regime, we always have $(\sqrt{d}/n\epsilon)^{1/2}<< (\sqrt{d}/n\epsilon)^{3/7}$.  When we state that $\epsilon$ is "sufficiently small," we refer to cases where the private term dominates the non-private term $n^{-1/3}$. We will clarify this in the revised version.

---

> > ### Comment · Reviewer_Yw5T · 2024-11-21
> >
> > Thank you for the clarifications! I have no further comments.

---

> > > ### Author Response · Authors · 2024-11-21
> > >
> > > Thank you for your response. Please don’t hesitate to reach out if you have any additional thoughts in the future.

---

### Official Review · Reviewer_3ujF · 2024-11-05

**Soundness:** 3
**Presentation:** 4
**Contribution:** 4
**Rating:** 8
**Confidence:** 4

**Summary:**

The authors find that finding second-order stationary points privately can be as fast as finding first-order stationary points privately at a rate of $\tilde{O} (\frac{1}{n^{\frac{1}{3}}} + (\frac{\sqrt{d}}{n\epsilon})^{\frac{1}{2}} )$.

**Strengths:**

1. The authors present in a nice and neat way.

2. This paper studies a timely and interesting problem - finding SOSP privately.

3. The authors introduced simple yet effective tools to solve this problem effectively. Tree mechanism has been applied in many DP papers but this paper illustrates its power when combined with the adaptive batch size. And the analysis part is non-trivial. This technique might be applicable to other private optimization problems.

**Weaknesses:**

I see no apparent weaknesses in this paper.

**Questions:**

Can the authors expand the related work section by briefly comparing this work to the momentum-based variance-reduction methods like [1] and
DP-SRM ([2])? I think these papers are somewhat relevant.

[1]: Tran, Hoang, and Ashok Cutkosky. "Momentum aggregation for private non-convex ERM." Advances in Neural Information Processing Systems 35 (2022): 10996-11008.

[2]: Wang, Lingxiao, et al. "Efficient privacy-preserving stochastic nonconvex optimization." Uncertainty in Artificial Intelligence. PMLR, 2023.

---

> ### Author Response · Authors · 2024-11-19
>
> Thank you for your positive feedback and for pointing out the related work that needs to be included. They are primarily about finding first-order stationary points; we will add them to the updated version.

---

### Meta-Review · Area_Chair_mTN4 · 2024-12-17

**Metareview:**

This paper uses adaptive batch sizes and incorporates the binary tree mechanism to improve the task of privately finding an SOSP. The derived bound matches the state-of-the-art in finding a FOSP, suggesting that privately finding a SOSP can be achieved at no additional cost.

The reviewers converged in praise of the novelty and significance of this theoretical work. The paper is well written and generally clear (after the discussion phase). All the reviewers provided positive scores and thus, I'm happy to recommend acceptance of this work with a spotlight presentation.

**Additional Comments On Reviewer Discussion:**

The authors took the reviewers' comments into consideration during the discussion phase. In particular, the authors improved the clarity of the work and provided more insights on sample complexity and data sharing.

---

### Decision · Program_Chairs · 2025-01-22

Accept (Spotlight)